# Generation of Brain Microvascular Endothelial-like Cells from Human iPS Cell-Derived Endothelial Progenitor Cells Using TGF-β Receptor Inhibitor, Laminin 511 Fragment, and Neuronal Cell Culture Supplements

**DOI:** 10.3390/pharmaceutics14122697

**Published:** 2022-12-02

**Authors:** Hiromasa Aoki, Misaki Yamashita, Tadahiro Hashita, Takahiro Iwao, Mineyoshi Aoyama, Tamihide Matsunaga

**Affiliations:** 1Department of Clinical Pharmacy, Graduate School of Pharmaceutical Sciences, Nagoya City University, Nagoya 467-8603, Japan; 2Department of Pathobiology, Graduate School of Pharmaceutical Sciences, Nagoya City University, Nagoya 467-8603, Japan

**Keywords:** human-induced pluripotent stem cell, endothelial progenitor cell, brain microvascular endothelial cell, laminin511, blood–brain barrier, transforming growth factor-β receptor inhibitor, B-27 supplement

## Abstract

Brain microvascular endothelial cells (BMECs) constitute the blood–brain barrier (BBB), which prevents the transfer of substances into the brain. Recently, in vitro BBB models using human-induced pluripotent stem (iPS) cell-derived brain microvascular endothelial-like cells (iBMELCs) have been created. However, it is suggested that iBMELCs differentiated by the existing methods are different from the BMECs that occur in vivo. This study aimed to establish iBMELCs generated via human iPS cell-derived endothelial progenitor cells (iEPCs) (E-iBMELCs). Expanded and cryopreserved iEPCs were thawed and differentiated into mature endothelial cells under various conditions. Intercellular barriers were significantly enhanced in E-iBMELCs using a B-27 supplement, transforming growth factor-β receptor inhibitor, and laminin 511 fragment. Expression of the endothelial cell markers was higher in the E-iBMELCs generated in this study compared with conventional methods. In addition, E-iBMELCs expressed P-glycoprotein. E-iBMELCs developed in this study will significantly contribute to drug discovery for neurodegenerative diseases and might elucidate the pathogenesis of neurodegenerative diseases associated with BBB disruption.

## 1. Introduction

Brain microvascular endothelial cells (BMECs), together with brain cells, such as astrocytes, pericytes, and neurons, constitute the blood–brain barrier (BBB). They are responsible for preventing the transfer of various substances, such as toxic molecules and cytokines, from the blood side to the brain side [1]. In many cases, the development of drug candidates that target the central nervous system fail to reach the brain parenchyma because they are prevented by the robust barrier function of the BBB. Therefore, the generation of high-quality and accurate in vitro evaluation systems for the BBB is important for the screening of therapeutic agents against neurodegenerative and BBB-related diseases. BMECs have specific features that distinguish them from peripheral vascular endothelial cells, including a robust intercellular adhesion structure (tight junction) and the expression of various transporters. Therefore, to establish robust BBB models, BMECs are needed with these features. Although human immortalized BMECs and rodent-derived BMECs are currently used, it is difficult to accurately evaluate the function of BBB using these systems because of low transendothelial electrical resistance (TEER) values, which are indicators of tight junction integrity [2] and species differences in humans [3]. In addition, human brain tissue is difficult to acquire, which limits the establishment of BBB models based on the primary human BMECs [4].

Recently, human-induced pluripotent stem (iPS) cell-derived brain microvascular endothelial-like cells (iBMELCs) were generated using relatively simple methods by Lippman et al. [5,6,7]. The iBMELCs developed by Lippman et al. (L-iBMELCs) are superior because they contain human-derived cells and have a high physical barrier function. However, the expression of endothelial markers was low in these L-iBMELCs, and its character as a vascular endothelial cell was considered weak in our previous study [8]. Recent reports have also shown that L-iBMELCs exhibit both vascular endothelial and epithelial properties, suggesting that they are different from in vivo BMECs [9]. In addition, because endothelial cells are not selected for differentiation steps into L-iBMECs, it is difficult to assess the quality of the differentiated cells. Moreover, there is no method to expand differentiated cells, and it is difficult to acquire a stable supply of L-iBMELCs.

Endothelial cells or endothelial progenitor cells invade the brain during embryogenesis, differentiate, and mature into BMECs by interacting with brain pericytes, astrocytes, and the extracellular matrix [10]. Therefore, we hypothesized that iBMELCs differentiated from human iPS cell-derived endothelial progenitor cells (iEPCs) (E-iBMELCs) have features similar to human BMECs in vivo. Previously, we succeeded in generating high-purity iEPCs by supplementing them with three small molecule compounds [Y-27632, a Rho kinase inhibitor; CHIR99021, a glycogen synthase kinase 3β inhibitor, and A-83-01, a transforming growth factor-β (TGF-β) receptor inhibitor], as well as by freeze-thawing the expanded iEPCs [11]. The aim of this study was to develop a method to generate iBMELCs, similar to BMECs in vivo, using iEPCs. The resulting E-iBMELCs exhibited TEER values of >100 Ω × cm^2^ from the expanded and cryopreserved iEPCs by culturing under a specific condition for several days. In addition, the permeability of substances by the paracellular pathway was significantly reduced compared with that in iEPCs. The results of immunofluorescence and gene expression analysis indicated that E-iBMELCs were highly similar to BMECs because of their high expression of vascular endothelial cell markers, tight junction proteins, and efflux transporters. Furthermore, since the cell source of this method is not human iPS cells, but rather iEPCs that have been cryopreserved after expanding in the culture, the cost and time of generating the cells is significantly reduced. Therefore, this method is expected to make a significant contribution to drug discovery and understanding the pathology associated with the BBB.

## 2. Methods

### 2.1. Materials

The E8 fragments of Laminin 211 (LN221F), laminin 411 (LN411F), and laminin 511 (LN511F) were purchased from Nippi, Inc. (Tokyo, Japan). Fibronectin and platelet-poor plasma-derived bovine serum (PDS) were procured from FUJIFILM WAKO (Osaka, Japan). Collagen type IV was purchased from Nitta Gelatin Inc. (Osaka, Japan). Human endothelial serum-free medium (HE-SFM), Essential 8 Flex medium, vitronectin (VTN-N), chemically defined lipid concentrate, 1,1′-dioctadecyl-3,3,3′,3′-tetramethyl-indocarbocyanine perchlorate acetylated low-density lipoprotein (Dil-Ac-LDL), 4-(2-hydroxyethyl)-1-piperazineethanesulfonic acid (HEPES) solution (1 M, pH 7.0–7.6), B-27, Hanks’ balanced salt solution with calcium and magnesium, and lucifer yellow (LY) were purchased from Thermo Fisher Scientific (Waltham, MA, USA). Endothelial Cell Basal Medium 2 was procured from Lonza (Basel, Switzerland). Cell culture inserts (1.0 µm transparent PET membrane) for a 12-well plate and Matrigel GFR were purchased from Corning, Inc. (Corning, NY, USA). Fetal bovine serum (FBS), FITC-dextran 4000 (FD4), and gelatin were procured from Sigma-Aldrich Corporation (St. Louis, MO, USA). TC Protector was purchased from DS Pharma Biomedical (Osaka, Japan). KnockOut Serum Replacement (KSR) was procured from Invitrogen (Carlsbad, CA, USA). A-83-01 was purchased from Cayman Chemical (Ann Arbor, MI, USA)

### 2.2. Cells

A human iPS cell line, 610B1, and an immortalized human BMEC cell line, hCMEC/D3, were procured from the Riken BioResource Center (Tsukuba, Japan) and Merck Millipore (Burlington, MA, USA), respectively.

### 2.3. Cell Culture

Cells were cultured based on our previously described method [11,12], and 610B1 cells were maintained in Essential 8 Flex medium on 1 μg/cm^2^ vitronectin (VTN-N)-coated dishes. In the case of L-iBMELCs, the human iPS cells were maintained on a feeder layer. hCMEC/D3 cells were cultured in Endothelial Cell Basal Medium 2 supplemented with 5% FBS, 5 μg/mL L-ascorbic acid phosphate magnesium salt n-hydrate, 1% chemically defined lipid concentrate, 10 mM HEPES solution, 1× penicillin-streptomycin solution, 1.4 μM hydrocortisone, and 1 ng/mL FGF2.

### 2.4. Coating before Differentiation

Gelatin was diluted with distilled water to 0.1% and autoclaved. The diluted solution was applied to cell culture dishes and incubated at 37 °C for 1 h or at 4 °C overnight. VTN-N was diluted with D-phosphate-buffered saline without Ca^2+^ and Mg^2+^ [D-PBS (−)] to 1 μg/cm^2^ and incubated on cell culture dishes at 37 °C for 1–2 h. Then, 0.01% collagen type I solution (Research Institute for the Functional Peptides Co., Ltd., Yamagata, Japan) was transferred to the culture dishes, aspirated, and left to dry. Fibronectin and collagen type IV were diluted with D-PBS (−) to 100 μg/mL and 400 μg/mL, respectively, and added to the well plates or cell culture inserts and incubated at 37 °C for 2–4 h. For three-component coating, VTN-N or the laminin fragments were added to the mixture of fibronectin and collagen type IV for coating. Matrigel GFR was diluted 1:30 in DMEM/F12 on ice. The diluted solution was transferred into 6-well plates with a cold tip and incubated at 37 °C for 1 h.

### 2.5. Differentiation of Human iPS Cells into iEPCs

iEPCs were differentiated, purified, and expanded based on our previously described method [11]. Here, we modified the method of expansion and purification. On day 11, the cells were passaged using TrypLE Select after washing with D-PBS (−). In case of the presence of extra cells, they were removed using the following steps. Extra cells were dissociated by tapping several times until they began to peel off from the dish after the addition of TrypLE Select. The cells were washed once with D-PBS (−) after aspirating off TrypLE Select containing extra cells. Then, the iEPCs were passaged using TrypLE Select.

### 2.6. Freeze-Thawing the Cultured Cells

The cell pellets (iEPCs on day 14 from the start of differentiation or hCMEC/D3 cells) were resuspended in TC Protector and stored in a deep freezer at −80 °C. The frozen cells were semi-thawed at 37 °C in a water bath and completely dissolved in prewarmed medium. Then, the cells were transferred to a 15 mL tube containing 10 mL of prewarmed medium and centrifuged at 100× *g* for 5 min. The obtained cell pellet was then resuspended in medium and seeded onto plates or cell culture inserts coated with 2 or 3 components.

### 2.7. Differentiation of iEPCs into E-iBMELCs

Thawed iEPCs were seeded on coated cell culture inserts (0.5–9 × 10^5^ cells/insert), 12-well plates (3.5 × 10^5^ cells/well), or 96-well plates (0.3 × 10^5^ cells/well). iEPCs were grown in HE-SFM containing 1 × penicillin-streptomycin solution and 20 ng/mL FGF2 supplemented with one of following factors: 2–10% KSR, 5% FBS, 5% PDS, 2–10% B-27, or 7.5% B-27 and 0.1–10 μM A-83-01. For the comparison of barrier function between iEPCs and hCMEC/D3 cells, hCMEC/D3 cells (passage 4) were cultured in the same way as iEPCs.

### 2.8. Differentiation of iBMELCs by a Modified Lippman’s Protocol

L-iBMELCs were differentiated using a previously reported method [12], which is a modification of the method reported by Lippman et al. [7]. The differentiated cells were seeded into 12-well type cell culture inserts (1.12 × 10^6^ cells/well) on day 8 and analyzed on day 10.

### 2.9. TEER Value Measurement

Before measurement, the cells were maintained for 15 min and brought to room temperature. The electrical resistance values were measured using a Millicell ERS-2 volt/ohm meter with a chopstick electrode (Merck Millipore) according to the manufacturer’s instructions. The electrical resistance values of the cell layer (R_cells_) were obtained by subtracting the electrical resistance values of only the membrane without cells (blank) from all measurements. TEER (Ω × cm^2^) was normalized by multiplying R_cells_ (Ω) and the membrane area (cm^2^). The membrane area of the 12-well type cell culture insert was set at 1.12 cm^2^.
$$TEER\;\left(\Omega \;\times {\;\mathrm{cm}}^{2}\right)=R_{cells}\;\left(\Omega \right)\times area\;{\left({\mathrm{cm}}^{2}\right)}^{\;}$$

### 2.10. Paracellular Permeability Assay

On day 4 after seeding iEPCs on cell culture inserts, the medium was replaced with a transport buffer, i.e., Hanks’ balanced salt solution with calcium and magnesium containing 10 mM HEPES solution. The volumes of the transport buffer in the apical and basal sides were 500 and 1500 μL, respectively. The transport buffer, along with 1 mg/mL of FD4 or 300 μM LY, was added to the apical side, whereas only the transport buffer was added to the basal side. The cells were cultured at 37 °C for 60 min. Then, 100 μL of the solution was collected from the basal side, and the collection time was adjusted such that each well was incubated for 1 h. Fluorescent FD4 (excitation; 498 nm, emission; 522 nm) or LY (excitation; 428 nm, emission; 536 nm) signals were measured using a Synergy HTX multimode plate reader and analyzed using Gen 5 data analysis software (BioTek Instruments, Inc., Winooski, VA, USA). The *P_app_* value of FD4 and LY were calculated using the following formula:$$P_{app}=\frac{dQ}{dt}\times \frac{1}{A\times C_{0}}$$
where *dQ/dt*, *A*, and C_0_ represent the amount of compound permeated per unit of time, surface area of the transport membrane, and initial compound concentration in the donor chamber, respectively.

### 2.11. RNA Extraction and qRT-PCR

Real-time PCR for mRNA quantitation was performed, and the same primers were used as those previously described [8,11,12]. Total RNA from human primary BMECs (hBMECs) was procured from ScienCell Research Laboratories, Inc. (Carlsbad, CA, USA). The cell lysates were obtained from cells cultured on 12-well plates. Total RNA was purified using the Agencourt RNAdvance Tissue Kit (Beckman Coulter, Inc., Brea, CA, USA) and reverse-transcribed using ReverTra Ace qPCR RT Master Mix (TOYOBO Co., Ltd., Osaka, Japan). The mRNA expression levels were measured using the KAPA SYBR^®^ FAST qPCR Kit (F. Hoffmann-La Roche, Ltd., Basel, Switzerland) with a LightCycler^®^ 96 System (F. Hoffmann-La Roche, Ltd.) and calculated using the 2^−ΔΔCT^ method. The values were normalized to the expression of hypoxanthine guanine phosphoribosyltransferase 1 (HPRT1).

### 2.12. Immunofluorescence Analysis

Immunofluorescence staining was performed, and the same antibodies were used as previously described [8,11,12].

### 2.13. Dil-Ac-LDL Uptake Assay

Differentiated cells in the wells of CellCarrier-96 microplates were incubated with 10 μg/mL Dil-Ac-LDL for 5 h, then washed 4 times with medium after incubating with 10 µg/mL Hoechst 33,342 at room temperature for 10 min. The samples were visualized using an Operetta High-Content Imaging System.

### 2.14. Matrigel Tube Formation Assay

Tube formation assay was performed as previously described [11,12].

### 2.15. Functional Analysis of P-glycoprotein (P-gp)

Permeability through P-gp was assessed as previously described [8,12].

### 2.16. Statistical Analysis

The differences in protein expression and accumulation used Rhodamin 123 comparisons between two groups using a two-tailed Student’s *t*-test. The comparison of TEER values measured after a prolonged period was assessed using two-way repeated measures analysis of variance. The comparison of three or more groups was assessed using a one-way functional analysis of variance with Tukey’s HSD or Games–Howell *post-hoc* test. SPSS Statistics version 25.0 (IBM Japan, Tokyo, Japan) was used for all statistical analyses, and significance was inferred *p* < 0.05.

## 3. Results

### 3.1. B-27 Supplement, TGF-β Receptor Inhibitor, and LN511F Markedly Improve the Physical Barrier Function of iEPCs

iEPCs were expanded and cryopreserved (Appendix A) based on our previous study [11]. The iEPCs, which were positive for PECAM1 and CD34, exhibited Dil-Ac-LDL uptake ability and tube-like structures (Appendix A), as shown previously [11]. We also previously demonstrated that the gene expression, tube-like structure formation, and Dil-Ac-LDL uptake in iEPCs were not affected by freeze-thawing [11]. In the present study, to analyze the effect of freeze-thawing on the barrier function of iEPCs, the difference in TEER values using frozen and non-frozen iEPCs was measured. We used frozen iEPCs because there was no significant difference between the two groups (Appendix A). We evaluated various conditions for increased TEER values using thawed iEPCs (Figure 1A) and found that the neural cell culture supplement B-27 [13,14] significantly increased the TEER values of iEPCs compared with other supplements (Figure 1B). The TEER values increased depending on the concentration of B-27 supplement, particularly at 7.5% and 10% (Figure 1C). In our previous study, we discovered that the small molecular compound A-83-01, which is a TGF-β receptor inhibitor, also increased TEER values. Therefore, we determined whether A-83-01 in the medium supplemented with 7.5% B-27 increased the TEER values. The results indicated that TEER values increased and were dependent upon the concentration of A-83-01, particularly 1 µM and 10 µM (Figure 1D). BMECs are often cultured on plates coated with fibronectin and collagen type IV [5,6,15], which are abundant in the BBB. However, in the BBB in vivo, laminins including LN411 and LN511 are also present in basement membranes, as well as fibronectin and collagen type IV. To generate a coating that is more similar to the biological BBB environment, we hypothesized that barrier function may be enhanced by mixing fibronectin and collagen type IV with different basement membrane components. As a result, adding LN511F to the coating significantly increased the TEER values of iEPCs compared with VTN-N, LN221F, and LN411F (Figure 1E). The increasing in TEER values by LN511F was highest at low concentrations (10 µg/mL), but was not dependent upon concentrations above 10 µg/mL (Figure 1F). Addition of a mixture of LN511F and LN411F to the coating did not increase the TEER values.

Next, we examined the optimal number of cells needed for seeding in the cell culture insert under the best condition with 7.5% B-27 supplement, 1 μM A-83-01, and LN511F. The results indicated that seeding iEPCs at 1 × 10^5^ cells/well resulted in higher TEER values (Figure 1G and Appendix A). Based on these results, the TEER values of iEPCs seeded at 1 × 10^5^ cells/well under a combination of all conditions were measured over a long period of time. The results indicated that 7.5% B-27 supplement, 1 μM A-83-01, and 10 μg/mL LN511F maintained higher TEER values for a longer period of time compared with the other groups (Figure 1H). Although the TEER value is a convenient index for simple measurements of the strength of tight junctions, it should be noted that the value varies significantly depending on the measurement device and temperature. Therefore, we tested the permeability of FD4 and LY, which are known to permeate through the paracellular pathway. In the results, the group consisting of B-27 supplement, A-83-01, and LN511F showed significantly lower permeability compared with the other groups (Figure 1I,J). The condition that resulted in the highest barrier function was defined as the BMEC-inducing condition, and the cells differentiated using this condition were designated as E-iBMELCs.

### 3.2. E-iBMELCs Are Similar to Primary BMECs

The expression and localization of BMEC-related proteins in iEPCs (cells cultured with 7.5% KSR) and E-iBMELCs were analyzed by immunofluorescence staining (Figure 2A). PECAM1 and cadherin 5 (CDH5, a.k.a. VE-cadherin), which are representative endothelial cell markers, were expressed on the membrane of both cells. The tight junction-related proteins ZO-1 and claudin-5 were expressed in both cells, but were more localized on the plasma membrane in E-iBMELCs. The efflux transporters P-gp and breast cancer resistant protein (BCRP) were also expressed in both cells. In addition, glucose transporter 1 (GLUT1), which is highly expressed in BMECs, was also expressed in both cells. Quantification of protein expression revealed that ZO-1, BCRP, and GLUT1 were significantly increased in E-iBMELCs (Figure 2B). The expression of BMEC-related genes in E-iBMELCs was compared with that of L-iBMELCs, hCMEC/D3 cells (human immortalized BMECs) and hBMECs (human primary BMECs) by qRT-PCR analysis (Figure 2C). The results indicated that the expression levels of PECAM1 and CDH5 mRNA in E-iBMELCs were significantly higher compared with those in L-iBMELCs and hCMEC/D3 cells, and were similar to that in hBMECs. Although the expression of ZO-1, occludin, BCRP, and GLUT1 mRNA in E-iBMELCs was similar or higher compared with that in other cells, multidrug resistance protein 1 (MDR1) (gene encoding P-gp) in E-iBMELCs was significantly lower compared with hCMEC/D3 cells. The expression of BMEC-related genes in hCMECs/D3 was relatively similar to that in hBMECs.

Immortalized cells are more convenient than other cells because of their stable supply; however, their barrier function is remarkably low. Therefore, we determined whether it is possible to increase the barrier function of hCMEC/D3 cells by changing the culture conditions to that optimized for E-iBMELCs. In the results, we found that barrier function was increased in hCMEC/D3 cells (Appendix A). However, the TEER values for hCMEC/D3 cells with the optimized conditions were lower compared with that of iEPCs without such conditions. The function of P-gp, which is an efflux transporter and essential for BBB function, was analyzed by a substrate accumulation assay (Figure 2D). The intercellular accumulation of rhodamine123 was significantly increased by treatment with the inhibitor, indicating that E-iBMELCs have P-gp function.

## 4. Discussion

In this study, we successfully generated cryopreserved iEPC-derived iBMELCs that were more similar to in vivo BMECs than L-iBMELCs. The TEER values for the frozen and non-frozen groups were unchanged. We previously found that gene and protein expression, proliferation capacity, Dil-Ac-LDL uptake, and tube-like formation of iEPCs were unchanged depending on the presence or absence of freezing [11]. These results suggest that iEPCs are largely unaffected by freeze-thawing and that they are a suitable source for differentiation into iBMELCs.

To differentiate iEPCs into iBMELCs, we first considered changing the composition of the culture medium. The iEPC differentiation and expansion culture medium were supplemented with KSR, which is serum substitute suitable for maintaining undifferentiated iPS cells. Therefore, we searched for a superior supplement to KSR and found that B-27 supplement significantly increased TEER values. Because B-27 supplement is essential for neuronal cell culture, we hypothesized that its addition to the culture medium would more closely mimic the environment of the brain. Considering that BMECs are differentiated from EPCs in the brain during embryogenesis, this is an expected result. Conversely, the addition of FBS significantly decreased the TEER values of iBMELCs, and PDS did not affect TEER values. These results indicate that the presence of various nutrients, hormones, and cytokines in FBS [16], particularly platelet-derived components, strongly affect iEPCs. Next, we determined whether a TGF-β receptor inhibitor, A-83-01, increases the TEER values of iBMELCs. We previously found that TGF-β receptor inhibitors significantly improve the barrier function of L-iBMELCs [12]. In the present study, A-83-01 increased the physical barrier function of iEPC-derived iBMELCs, suggesting that the regulation of TGF-β function is involved in the acquisition and maintenance of a strong BBB function. Finally, we determined whether the extracellular matrix affects the TEER values of iBMELCs. The BBB is a basement membrane-rich structure, and the basement membrane is considered to impart a strong barrier function to the BBB [17]. Although fibronectin and collagen type Ⅳ are primarily used for iBMELC cultures in cell culture inserts [5,18], laminins, such as LN411 and LN511, are also contained in basement membrane components of the BBB in vivo. Therefore, we added each laminin in combination with fibronectin and collagen type IV and found that the addition of LN511F significantly increased the TEER values. We previously reported that L-iBMELCs cultured on LN221F from early to mid-differentiation exhibited high TEER values [8]; however, in this study, laminin 221 did not affect the TEER values of iEPC-derived iBMELCs. This may be the result of the different differentiation methods used for L-iBMELCs and iEPC-derived iBMELCs. Based on previous studies reporting that laminin 511 is more abundant than laminin 221 in BMECs [19], we considered the results of this study that laminin 511 has a greater contribution to barrier function to be reasonable. In addition, LN511F alone increased TEER values to a lesser degree compared with a mixture of collagen type Ⅳ and fibronectin, suggesting that a mixture of collagen type IV and fibronectin is necessary to increase physical barrier function. Recently, it was reported that LN511F alone increases TEER values compared with fibronectin and collagen type Ⅳ in L-iBMELCs [20]; however, LN511F alone has also been shown to decrease gene the expression of endothelial cell markers. The increase in TEER values may be the result of LN511F orienting cells toward non-endothelial cells.

A comparison of the expression levels of BMEC markers in iEPCs (KSR group) and E-iBMELCs (B-27, A-83-01, LN511F group) revealed a marked increase in the ZO-1, BCRP, and GLUT1 protein expression. A slight upregulation of claudin-5 and P-gp was also observed, and claudin-5 was more strongly localized to the cell membrane. These results suggest that the physical barrier function of E-iBMELCs is increased compared with iEPCs and that E-iBMELCs have acquired the properties of in vivo BMECs. Although the L-iBMELCs show weak membrane localization for the vascular endothelial cell markers, PECAM1 and CDH5 [8,9,12,21], E-iBMELCs produced by the present method exhibited robust localization of PECAM1 and CDH5 on the plasma membrane, similar to that of iEPCs. In addition, the expressions of PECAM1 and CDH5 mRNA in L-iBMELCs were remarkably low, whereas that in the E-iBMELCs generated by the present method were close to that in hBMECs. These results indicate that E-iBMELCs produced by our method exhibit strong characteristics as vascular endothelial cells, unlike the L-iBMELCs. The gene expression levels of ZO-1, occludin, GLUT1, and BCRP in the E-iBMELCs produced by this method were higher than those in hBMECs and hCMEC/D3 cells. Although the expression of the MDR1 gene in E-iBMELCs was lower compared with that in hCMEC/D3 cells, the levels were almost the same as that of hBMECs. Furthermore, E-iBMELCs exhibited P-gp function, which is responsible for preventing the influx of toxic substances into the brain parenchyma. This suggests that they differentiated into mature endothelial cells that are very similar to BMECs in vivo.

The problem with hCMEC/D3 cells, which are immortalized cells and therefore, infinitely proliferating and easy to generate, is that they have low TEER values. Thus, we determined whether the TEER values of hCMEC/D3 cells were increased by the BMEC-inducing condition containing B-27, A-83-01, and LN511F, and found that the TEER values were significantly increased. These results suggest that our BMEC-inducing condition is versatile and increases TEER; however, the TEER value was lower compared with that of iEPCs without the BMEC-inducing condition. Therefore, the development of another method was needed to yield the same properties as BMECs in vivo as immortalized cells. Recently, Praca et al. reported that iBMELCs could be generated from human iPS cells that were derived from vascular endothelial progenitor cells using VEGF, Wnt3a, and retinoic acid [22]. Although the cells were similar to in vivo BMECs based on various analyses, the problems included low TEER values (approximately 60 Ω × cm^2^) and the lack of a physical barrier function (LY permeability; approximately 1 × 10^−3^ cm/min). Compared with iBMELCs reported by Praca et al., E-iBMELCs in this study have a barrier function closer to that in vivo, because they exhibit higher TEER values (above 100 Ω × cm^2^) and lower paracellular permeability [LY permeability; approximately 4 × 10^−6^ cm/s (2.4 × 10^−4^ cm/min)]. In contrast, the L-iBMELCs had high TEER values (above 1000 Ω × cm^2^) without a co-culture system, which is close to the barrier function of the BBB in vivo [23,24], although the previous studies suggest that L-iBMECs differ from in vivo BMECs [9,25]. Consequently, we succeeded in generating cryopreserved iEPC-derived iBMELCs, which retain robust tight junction function compared with previous E-iBMELCs and exhibit the properties of endothelial cells that are not present in L-iBMELCs. However, the TEER values of E-iBMELCs remain low and further studies are needed to complete an in vitro BBB model that mimics living organisms, using this study as a stepping stone.

## 5. Conclusions

The iBMELCs differentiated from iEPCs using a B-27 supplement, TGF-β receptor inhibitor, and LN511F exhibited the strong characteristics of vascular endothelial cells and a higher barrier function. In addition, these cells expressed BMEC markers and P-gp function, suggesting that their properties are very similar to those of in vivo BMECs, which were superior to cells produced previously using differentiation methods. Furthermore, iEPCs are a readily accessible cell source because they can be expanded and cryopreserved. In conclusion, E-iBMELCs produced in this study may be applied to a variety of studies, including drug discovery targeting the central nervous system, elucidating the pathogenesis of neurodegenerative diseases associated with BBB disruption and differentiation associated with the BBB.

## Figures and Tables

**Figure 1 pharmaceutics-14-02697-f001:**
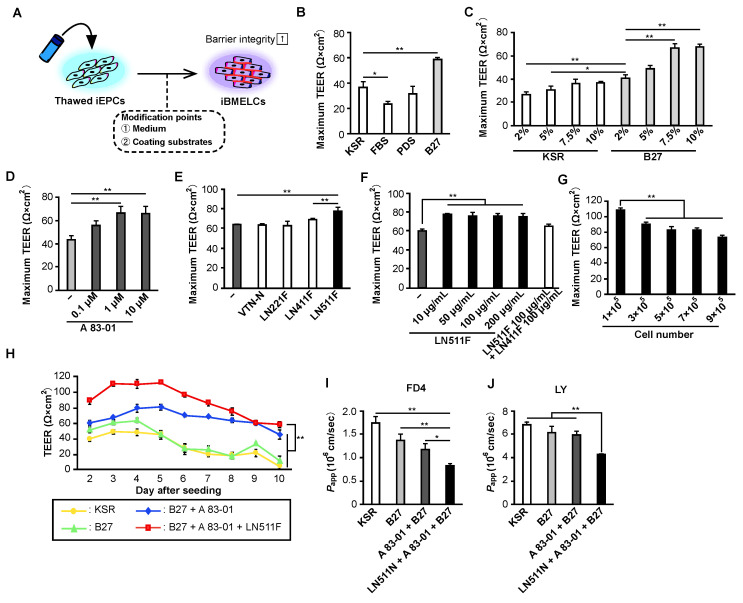
Evaluation of the physical barrier function of iEPCs under various conditions. (**A**) Strategy for the differentiation of E-iBMELCs. (**B**–**F**) Thawed iEPCs (7 × 10^5^ cells/insert) were cultured under the indicated conditions. Measurement of TEER values from day 2 to 10 and the maximum value among them is shown. Data are presented as the mean ± SD (*n* = 3; * *p* < 0.05, ** *p* < 0.01). (**B**) Tukey’s HSD test; KSR group vs. others. (**C**) Tukey’s HSD test; 2% B-27 group vs. others. (**D**) Tukey’s HSD test; control vs. others. (**E**) Tukey’s HSD test; control vs. others, LN411F group vs. LN511F group. (**F**) Tukey’s HSD test; control vs. others. (**G**) Thawed iEPCs were seeded at the indicated number of cells and cultured under the indicated conditions. Measurement of TEER values of iEPCs from day 2 to 10 and the maximum value among them is shown. Data are presented as the mean ± SD (*n* = 3; ** *p* < 0.01; Tukey’s HSD test; 1 × 10^5^ group vs. others). (**H**) Measurement of TEER values of iEPCs (1 × 10^5^ cells/insert) cultured under the indicated conditions from day 2 to 10. Data are presented as the mean ± SD (*n* = 3; ** *p* < 0.01; two-way repeated measures analysis of variance; B-27, A-83-01, and LN511F group vs. others). (**I**,**J**) FD4 and LY permeability assay of iEPCs (1 × 10^5^ cells/insert) cultured under the indicated conditions on day 4. *P_app_*, apparent permeability coefficient. Data are presented as the mean ± SD (*n* = 3; * *p* < 0.05, ** *p* < 0.01; Tukey’s HSD test; B-27, A-83-01, and LN511F group vs. others).

**Figure 2 pharmaceutics-14-02697-f002:**
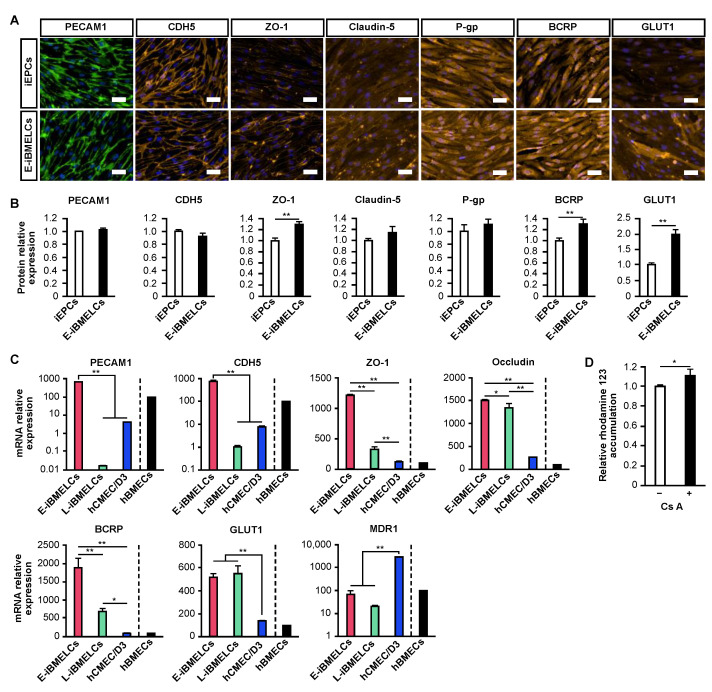
Characterization of E-iBMELCs. (**A**,**B**) Immunofluorescence analysis of PECAM1 (green), CDH5 (orange), ZO-1 (orange), claudin-5 (orange), P-gp (orange), BCRP (orange), and GLUT1 (orange) expression in iEPCs (cultured in HE-SFM supplemented with 20 ng/mL FGF2, penicillin-streptomycin solution, 7.5% KSR on a cell culture insert coated with a coating solution consisting of fibronectin and collagen type IV) and E-iBMELCs (cultured in HE-SFM supplemented with 20 ng/mL FGF2, penicillin-streptomycin solution, 7.5% B-27 supplement, and 1 μM A-83-01 on a cell culture insert coated with a coating solution consisting of fibronectin, collagen type IV and LN511F) on day 4. DAPI = blue. Scale bars = 50 μm. Relative protein expression levels were calculated based on fluorescence intensities. Data are presented as the mean ± SD (*n* = 3; ** *p* < 0.01; Student’s *t*-test). (**C**) Relative mRNA expression levels of PECAM1, CDH5, ZO-1, occludin, BCRP, GLUT1, and MDR1 in iBMELCs, L-iBMELCs, hCMEC/D3 cells, and hBMECs. The values are normalized to the expression of HPRT1. The relative mRNA expression levels of hBMECs were defined as 100. Data are presented as the mean ± SD (*n* = 3; * *p* < 0.05, ** *p* < 0.01; Tukey’s HSD test). hBMECs: *n* = 1. (**D**) Relative intracellular accumulation of rhodamine 123. The E-iBMELCs on day 4 were incubated with 10 μM rhodamine 123 in the absence or presence of 10 μM cyclosporin A (CsA) for 1 h at 37 °C. Relative fluorescence intensity values were normalized to a condition without CsA (set to 1). Data are presented as the mean ± SD (*n* = 3; * *p* < 0.05; Student’s *t*-test).

## Data Availability

The datasets used and/or analyzed in the current study are available from the corresponding author upon reasonable request.

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
