# Peer review of "Generation of Brain Microvascular Endothelial-like Cells from Human iPS Cell-Derived Endothelial Progenitor Cells Using TGF-β Receptor Inhibitor, Laminin 511 Fragment, and Neuronal Cell Culture Supplements"

_pharmaceutics, 2022, doi:10.3390/pharmaceutics14122697_

Round 1

Reviewer 1 Report

Journal: Pharmaceutics

Manuscript ID: pharmaceutics-2045697

Title: Generation of brain microvascular endothelial-like cells from 2 human iPS cell-derived endothelial progenitor cells using the 3 TGF-β receptor inhibitor, laminin 511 fragment, and neuronal 4 cell culture supplements

Dear Editors,

In the current study, the Authors aimed to establish induced pluripotent stem (iPS) cell-derived brain microvascular endothelial-like cells (iBMELCs) generated via human iPS cell-derived endothelial progenitor cells (iEPCs) (E-iBMELCs). For this purpose, they differentiated iEPCs into mature endothelial cells under various in vitro conditions and found that intercellular barriers were significantly enhanced in E-iBMELCs using a B-27 supplement, TGF-β receptor inhibitor, and laminin 511 fragment. They also observed that expression of the endothelial cell markers was higher in the E-iBMELCs generated in this study compared with conventional methods, and importantly they found that E-iBMELCs expressed P-glycoprotein. They concluded that E-iBMELCs that they have established can be used as an intro BBB model to significantly contribute to drug discovery for neurodegenerative diseases and to elucidate the pathogenesis of neurodegenerative diseases associated with BBB disruption.

The findings mentioned above are interesting, and the manuscript has been written logically, with a satisfactory level of profession. Moreover, the methodology is sound. In the following, certain minor comments are directed. In conclusion, the mentioned paper deserves to be published in this Journal provided that a satisfactory minor revision is undertaken.

Kind regards,

1. All abbreviations in the text (e.g. Vtn-n, and the meaning of the initial letters of L-iBMELCs and E-iBMELCs, etc.) should be checked and they should be written in full upon their first appearance.

2. The language and the grammatical phrase construction specifically in the “Introduction” section should be improved.

3. In line 39, the phrase “… and to elucidate of…” should be corrected as “… and to elucidate …”.

4. In line 63, the phrase “human induced pluripotent stem cells (iPS) cell-derived …” should be corrected as “human induced pluripotent stem (iPS) cell-derived …”.

5. In line 68, the phrase “… in our previously study” should be corrected as “… in our previous study”.

6. The Authors may choose to focus more on their aims rather than their findings and conclusions in the last paragraph of the “Introduction” section.

7. In “2.11 Immunofluorescence analysis” and “2.14 Functional analysis of P-glycoprotein (P-gp)” subsections of the “Methods” section, a brief outline of the methodology is needed.

8. The second paragraph of the “3.2 E-iBMELCs are similar to primary BMECs” subsection of the “Results” section is structured as a discussion rather than of statement of data, therefore the Authors should consider moving the mentioned paragraph to the “Discussion” section, or rewriting it.

9. In lines 401 and 402, the phrase “… a previous study reported that laminin 221 is not the basement membrane of the BBB, whereas laminin 511 is abundant” should be revised.

10. In line 462, the phrase “… are may be applied to …” should be corrected as “… may be applied to …”.

Reviewer 2 Report

Very nice work. Some comments need to be addressed.

1. As declared, TEERS are not a good enough indicator of the BBB properties in culture. Then, the authors analyze cell permeability for FD4 and LY. However, I suggest analyzing the cell permeability using high molecular-weight fluorescent tracers.  

2. Clarify how the mRNA expression levels were calculated. Differences of 1000-2000 folds in some of the markers between hCMEC/D3 and E/iBMELCs are impressive. Please clarify.

3. E-iBMELCs seem a good model. Then, any change to challenge with a stimulus that disrupts the BBB model?
